# Fine scale human mobility changes within 26 US cities in 2020 in response to the COVID-19 pandemic were associated with distance and income

Rohan Arambepola[1]*, Kathryn L. Schaber[1], Catherine Schluth[1], Angkana T. Huang[2], Alain B. Labrique[3], Shruti H. Mehta[1], Sunil S. Solomon[1,4], Derek A. T. Cummings[5], Amy Wesolowski[1]

1 Department of Epidemiology, Johns Hopkins Bloomberg School of Public Health, Baltimore, MD, United States of America, 2 Department of Genetics, Cambridge University, Cambridge, United Kingdom, 3 Department of International Health, Johns Hopkins Bloomberg School of Public Health, Baltimore, MD, United States of America, 4 Department of Infectious Diseases, Johns Hopkins School of Medicine, Baltimore, MD, United States of America, 5 Department of Biology and the Emerging Pathogens Institute, University of Florida, Gainesville, FL, United States of America

* rarambe1@jh.edu

**Data Availability Statement:** Data and code to replicate the analysis is available at https://github.com/rarambepola/US-cities-COVID-mobility. All

## Abstract

Human mobility patterns changed greatly due to the COVID-19 pandemic. Despite many analyses investigating general mobility trends, there has been less work characterising changes in mobility on a fine spatial scale and developing frameworks to model these changes. We analyse zip code-level within-city mobility data from 26 US cities between February 2 –August 31, 2020. We use Bayesian models to characterise the initial decrease in mobility and mobility patterns between June—August at this fine spatial scale. There were similar temporal trends across cities but large variations in the magnitude of mobility reductions. Long-distance routes and higher-income subscribers, but not age, were associated with greater mobility reductions. At the city level, mobility rates around early April, when mobility was lowest, and over summer showed little association with non-pharmaceutical interventions or case rates. Changes in mobility patterns lasted until the end of the study period, despite overall numbers of trips recovering to near baseline levels in many cities.

## Introduction

The onset of the COVID-19 pandemic prompted rapid changes in human mobility around the world, driven by national and subnational travel restrictions, international border closures, self-isolation and quarantining, school and work closures, and individual attempts to limit numbers of social contacts [1–10]. Many analyses have investigated changes in overall mobility and the effects of mobility-based mitigation measures, often linking stay at home measures and lockdown restrictions to reductions in transmission [11–17]. Other work has focused on the effect of subnational movement restrictions on the time taken for transmission to be introduced in different locations [18, 19]. Stay-at-home orders, lockdowns, and national or

other data used has been referenced and is publicly available.

**Funding:** AW is supported by a Career Award at the Scientific Interface from the Burroughs Wellcome Fund. RA, KLS, CS and AW are all supported by National Institute of Health Director's New Innovator Award, grant number DP2LM013102-0. RA, KLS, CS, DATC, and AW are also supported by the National Institute of Allergy and Infectious Diseases (1R01A1160780-01). The funders had no role in study design, data collection and analysis, decision to publish, or preparation of the manuscript.

**Competing interests:** The authors have declared that no competing interests exist.

subnational border closures represent strict measures limiting mobility. More subtle changes in human mobility, whether in response to certain non-pharmaceutical interventions (NPIs) or perceived risk, may also have important implications for modelling and forecasting transmission. A more direct characterisation of the relationship between policy, epidemiological risk, and behaviour is needed to fully interpret the effect of mitigation measures and predict patterns of transmission. Many of the large agent-based transmission models that have guided public health policy, such as those used in the US and Europe, account for work-based contact structure but otherwise assume relative reductions in mobility or transmission rates that are largely homogeneous (over space and the population) when forecasting the effect of NPIs [20–23]. A better understanding of the heterogeneity in the changes in mobility that occurred in response to the pandemic–such as the types of journeys that decreased the most, the locations where individuals were able to limit travel, and how long these changes lasted–could improve these assumptions. Characterising mobility patterns in the absence of many NPIs is also likely to be important for forecasting transmission and identifying populations and regions most at risk going forward, as many countries lift their remaining mobility-based mitigation measures [24].

Data sets on human mobility have allowed travel to be characterized in the context of infectious disease epidemiology [25–27]. Mobile device data is a uniquely rich source of information on human movement, allowing the collection of mobility data on a much larger scale than travel surveys, for example, and at higher temporal and spatial resolutions than censuses [28–30]. Prior to the COVID-19 pandemic, the primary form of these data was Call Detail Records (CDRs), which detail the times and approximate locations of subscribers when devices are in use. These records are typically aggregated over the subscriber base, time, and space to obtain origin-destination matrices, which describe the flows of movement between different geographical units in each time frame [28, 29]. CDRs have been used to parameterise metapopulation models of COVID-19 transmission [19, 31] and quantify the extent to which mobility is associated with or drives transmission in specific locations [10, 32, 33]. However, there has been limited work using these data to characterise and provide generalisable insights into how the structure of mobility patterns changed over the course of the pandemic. One example of this kind of analysis is the work of Schlosser et al., who found reductions in long-distance travel in Germany that lasted into the summer of 2020 [34]. Kishore et al. also analysed mobile device data in several countries and found that lockdown announcements in early 2020 led to increased urban-to-rural migration [35]. Mobile device data has the potential to provide an unprecedented level of detail on temporal patterns of mobility during the COVID-19 pandemic. However, to date few approaches have leveraged theses sources to develop modelling frameworks that capture the heterogeneity in observed travel. Furthermore, few analyses have taken advantage of the ability of mobile device data to provide information on human mobility at a fine spatial scale.

Here we analyse travel between zip codes within 26 cities in the US between February 2 – August 31, 2020, quantified using mobile device data. We primarily focus on travel over two distinct time periods–the initial onset of the COVID-19 pandemic in the US and summer 2020 –and investigate how mobility at this fine spatial scale changed during these periods and the demographic, epidemiological, and policy factors associated with these changes.

## Methods

### Ethics statement

The study was approved by the Institutional Review Board of Johns Hopkins Bloomberg School of Public Health (IRB00015892). The consent for participants was waived by the

Institutional Review Board of Johns Hopkins Bloomberg School of Public Health. All methods were carried out in accordance with relevant guidelines and regulations.

## Study area

In an effort to help study COVID-19 transmission and associated mobility early in the pandemic, a large U.S. mobile phone operator supplied non-identifiable, aggregated data about movement patterns of subscribers between zip codes in 26 cities in the US. These cities were chosen in collaboration with the mobile phone operator based on where sufficient data was available and with the aim of including cities across the country that were, at the time, experiencing different transmission dynamics.

Zip codes do not correspond to a specific administrative unit but are small (lower than administrative level 2 with around 14 zip codes per admin 2 unit on average) in both area (average area of 80 square miles) and population (average of 6,000 people). For each city, first all zip codes within the metro area or within 20 miles of the city centre were included. Then for each zip code included in the previous step, all zip codes in the same county were also included. To aid interpretation of the results, the cities were grouped into six informal geographic regions (listed in Table 1). The locations of the cities are shown in Fig 1A. Full lists of the zip codes included for each city can be found in S1 Data.

**Table 1.  Information on the cities in the dataset.**

| City | State | Region | Number of zip codes |
|---|---|---|---|
| Baltimore | Maryland | Northeast | 283 |
| New York | New York | Northeast | 1074 |
| Philadelphia | Pennsylvania | Northeast | 367 |
| Atlanta | Georgia | Southeast | 219 |
| Charlotte | North Carolina | Southeast | 106 |
| Jacksonville | Florida | Southeast | 62 |
| Miami | Florida | Southeast | 187 |
| Nashville | Tennessee | Southeast | 121 |
| Tampa | Florida | Southeast | 155 |
| Austin | Texas | South | 89 |
| Dallas | Texas | South | 285 |
| El Paso | Texas | South | 78 |
| Houston | Texas | South | 253 |
| Phoenix | Arizona | South | 164 |
| San Antonio | Texas | South | 118 |
| Los Angeles | California | West coast | 389 |
| San Diego | California | West coast | 114 |
| San Francisco | California | West coast | 195 |
| San Jose | California | West coast | 161 |
| Chicago | Illinois | Great lakes | 387 |
| Columbus | Ohio | Great lakes | 130 |
| Detroit | Michigan | Great lakes | 262 |
| Fargo | North Dakota | Midwest | 70 |
| Lincoln | Nebraska | Midwest | 106 |
| Omaha | Nebraska | Midwest | 117 |
| Sioux Falls | South Dakota | Midwest | 56 |

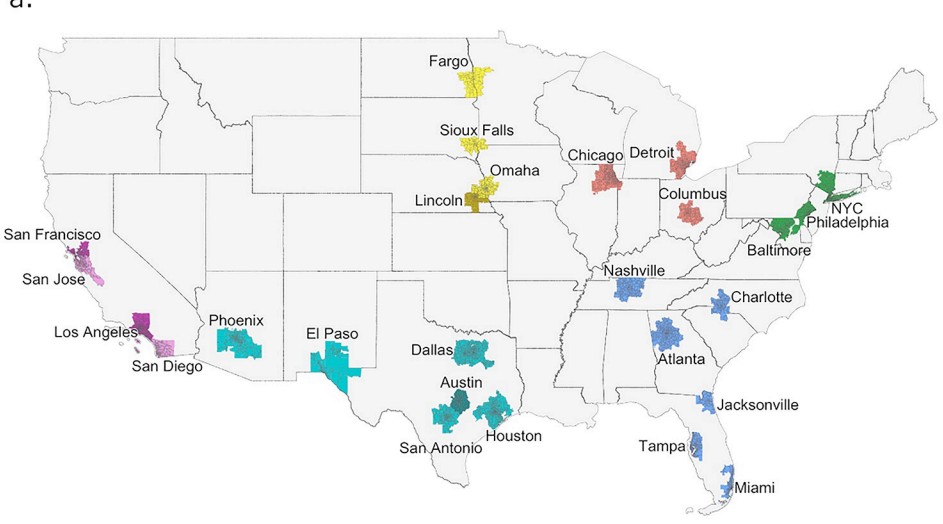

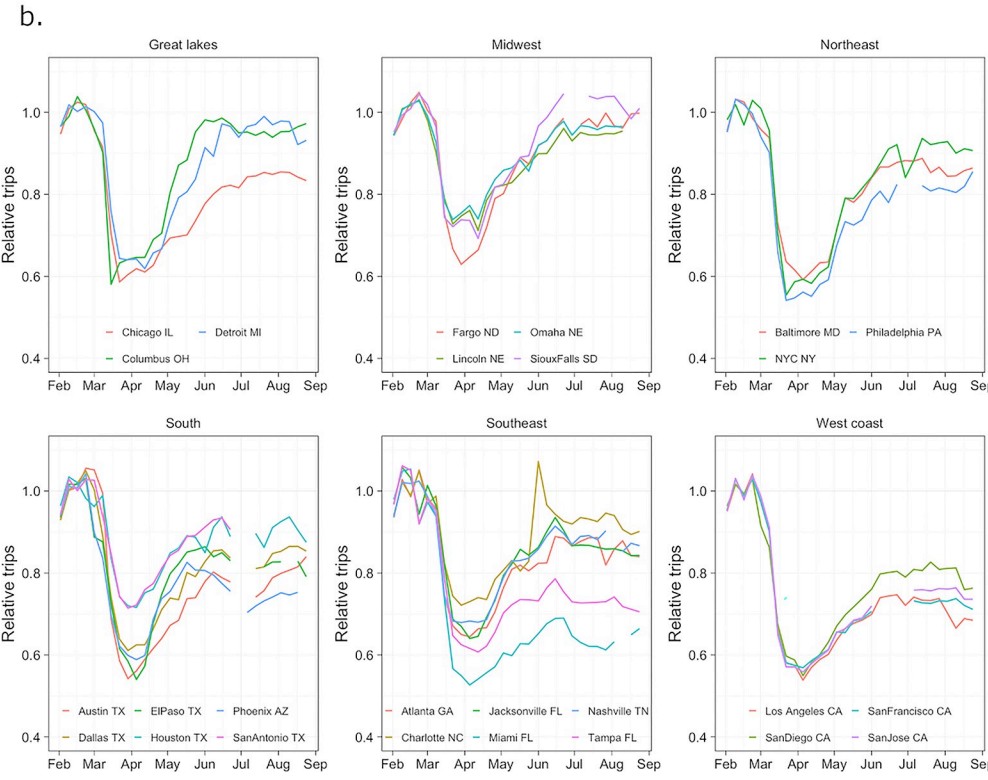

**Fig 1. Location of cities in the dataset and weekly travel in each city over time relative to baseline.** (a) Cities in dataset coloured by region. The base map created using a US Census Bureau shapefile [59]. (b) Weekly trips in each city (across all ages) over time relative to baseline travel in February. Mobility decreases sharply at the beginning of March in all cities, reaching a minimum around the beginning of February, and then increasing until early June where it levels off. Magnitude of reductions in mobility and levels after June are highly variable between cities, those generally more consistent within regions other than the South and Southeast. Weeks in which data loss occurred are not shown.

## Mobile device data

Daily aggregate data was obtained on the total number of trips taken by subscribers between any two zip codes within the same city. A trip was defined as a subscriber being registered as being in one zip code for at least 10 minutes and then subsequently in a different zip code for at least 10 minutes. Subscribers were split into three age groups (18–34, 35–54 and 55+ years) and data were provided as daily origin-destination matrices for each city and age group. Any values below 50 were not supplied to protect subscriber anonymity. Daily origin-destination matrices were aggregated each week for analysis. Data were available from February 2 to August 31, 2020. Some weeks in certain cities were excluded from the analysis due to data loss during the collection process (see S2 Data), which was identified by the mobile phone operator. This data was treated as missing at random. No data imputation was performed.

## Sampling and demographic data

Data was provided by the mobile phone operator where available but cannot be considered a random sample of the population. Demographic information was used to account for some of the bias introduced by this non-random sampling method. Aggregated data on the age and annual income of subscribers based in each zip code were obtained where available. Annual income was categorised into three groups: less than $25,000, $25,000 - $50,000, more than $50,000. As with travel information, sociodemographic information for groups with fewer than 50 members was not included to protect the privacy of subscribers. Information on the total population and median household income in each zip code was extracted from the results of the 2019 American Community Survey (ACS) [36].

## Case data

Daily recorded COVID-19 rates at the county-level were obtained from the COVID-19 Data Repository by the Center for Systems Science and Engineering (CSSE) at Johns Hopkins University using the COVIDcast Epidata API [37, 38]. Each zip code was associated with the case rate of the county it was in.

## Non-pharmaceutical interventions

Daily state-level data on the stringency of NPIs in effect over the study period were obtained from the Oxford COVID-19 Government Response Tracker [39, 40]. The stringency index is an average of indicators including school closures, workplace closures, cancellation of public events, national and international travel restrictions and stay at home orders. Data on the start and end of county-level stay-at-home/shelter-in-place orders and non-essential business closures were taken from the Keystone COVID-19 Intervention Dataset [41, 42] and county-level school closures from the Hikma Health COVID-19 Policy Dataset [43].

## Baseline travel rates

In order to compare mobility over time period to "normal" pre-pandemic levels, the month of February (the first four weeks of data) was taken as a baseline period. Relative mobility (for entire cities or subsets of zip codes) was calculated by dividing the number of trips in the week of interest by the average number of weekly trips during the baseline period.

## The effect of trip distance on mobility

For each city, trip distance was categorised into four quartiles, defined with respect to the number of trips made of different distances in that city across the entire study period. The

distance between two zip codes was defined as the distance between their centroids. These distance categories were used in the Bayesian models of travel (see below) and equivalent results where the distance quartiles were based only on trips made during February 2020 are included in the S1 Text.

## The effect of subscriber income on mobility

Due to limited data available on subscribers in the less than $25,000 annual income category (see S2 Text), this group was combined with the $25,000–50,000 category for the analysis. Therefore, the final income groups for the purpose of analysis were subscribers with annual incomes up to $50,000 and above $50,000, which we refer to as lower-income and higher-income subscribers, respectively. To compare the rates of travel based on these groups, zip codes in each city were classified into two groups based on whether the proportion of higher-income subscribers was above or below the median for the city. Numbers of outward trips from zip codes in each group were then calculated for each city and week relative to the baseline period. The ratio of relative trips in zip codes with above and below median higher-income subscribers was then calculated for each week and city. A value of this ratio below one represents less travel relative to baseline in zip codes with greater proportions of higher-income subscribers compared to zip codes with lower proportions of higher-income subscribers.

## Modelling mobility in the initial months of the pandemic

A Bayesian model was developed to model the rate of reduction in trips on different routes over the initial 9 weeks of data (February 1 –April 4, 2020). Mobility in each city was modelled separately, with the model learning a baseline trip rate for each route (pair of zip codes) and weekly city-wide decreases in mobility that were modified by the effect of route- or zip code-specific explanatory variables. This allowed us to infer the effect of these explanatory variables and age on the rate of decrease in travel. Age was treated differently to other explanatory variables as the data was disaggregated by age.

In more detail, the rate of trips from zip code $i$ to zip code $j$ in week $t$ in age group $a$, $\lambda_{ijat}$, was modelled as

$$\lambda_{ijat} = r_{ija} \, q_{ijat}$$

where $r_{ija}$ was the learned baseline rate of travel for this pair of zip codes and age group and the fraction $q_{ijat}$ was modelled as the city-wide cumulative weekly decreases in travel modified by the effect of covariates. That is,

$$\log q_{ijat} = -e^{\beta^T X_{ijt} + \beta_a} \sum_{k=1}^{t} c_k$$

where $X_{ijt}$ were explanatory variables (associated with the zip code $i$ or the route) and the parameters $c_k > 0$ (city-wide weekly decreases in travel), $\beta$ (effect of explanatory variables), and $\beta_a$ (effect of age group $a$) were learned. The explanatory variables were route distance quartile, proportion of higher-income subscribers, relative case rates (compared to the city) and median household income (for the whole zip code, not restricted to subscribers).

To connect this model structure to our data, we first note that while our data contained information on the direction of trips, it was not possible to associate each trip with a home zip code. An observed trip from zip code $i$ to zip code $j$ in a given week could therefore be an outward trip by an individual living in zip code $i$, a returning trip by an individual living in zip

code $j$, or part of an indirect journey made by an individual who lives in neither. Assuming that the volume of these third kinds of trips was generally insignificant relative to the first two, we modelled the number of trips between zip codes $i$ and $j$ (in either direction) in week $t$, $Y_{ijat}$, as the sum of independent processes (conditional on the underlying rates) in each direction, yielding the likelihood

$$Y_{ijat} \sim Poisson(p_{ia}\lambda_{ijat} + p_{ja}\lambda_{jiat})$$

where $p_{ia}$ was the number of subscribers in age group $a$ in zip code $i$. The model was completed by including suitable prior distributions for each parameter (see S3 Text). Note that the likelihood was slightly modified for the observations below 50 (which were censored), detailed in S4 Text.

To compare the effectiveness of this model for understanding factors influencing declining mobility with a common model of mobility, we also fit a model in which the baseline trip rates were based on a gravity model. That is, instead of the baseline rate for each route, $r_{ija}$, being a parameter to be learned, this was modelled as

$$r_{ija} = \theta \frac{p_{ia}{}^{\alpha} \bar{p}_j{}^{\delta}}{d(i,j)^{\gamma}}.$$

Here $\bar{p}_J$ was the total population of zip code $j$ according to the ACS (not the subscriber population), as this term is typically interpreted as a proxy for attractiveness of the destination location. $d(i,j)$ was the distance between zip codes $i$ and $j$, and $\alpha, \delta, \gamma, \theta$ were parameters to be learned.

## Modelling mobility over summer

To understand longer term changes in mobility, a second model was developed to investigate the relationship between mobility in a given time frame and baseline travel. This used the baseline trip rates learned in the previous model and modelled the trip rate in the time frame of interest as the product of this baseline rate and the effect of explanatory variables. The trip rate between zip codes $i$ and $j$ in age group $a$ during the time period of interest, $\lambda_{ija}$, was therefore modelled as

$$\lambda_{ija} = n_{weeks} e^{-(\beta_0 + \beta^T X_{ij} + \beta_a)} r_{ija}$$

where $n_{weeks}$ was the duration of the time period (in weeks) that was being compared to baseline rates, $X_{ij}$ were explanatory variables, and $\beta_0, \beta, \beta_a$ were parameters to be learned. The explanatory variables were route distance quartile, proportion of higher-income subscribers, and median household income. Again, prior distributions for these parameters are described in the S3 Text. This model was applied to travel between June 1 –August 31, 2020 to investigate mobility patterns in the summer. This period was chosen as increases in mobility rates had largely ended by the beginning of June and remained at similar levels throughout the rest of the study period. Weeks with data loss were not included for the relevant cities.

All models were fit using the R programming language and TMB package, using a Laplace approximation to the posterior distribution [44, 45]. A summary and comparison of the models is included in S1 Table.

## Results

Subscribers in the dataset with available income and age data over-represented the higher income and older age groups as compared to US census data on the general adult population, see S2 Text more details. The share of subscribers in each city out of all subscribers roughly

correlated with the size of each city by population, with the most subscribers in New York (13.3%) and least in Fargo (0.48%) (see S2 Table). There were at least 5,000 subscribers in each city.

## City-level mobility trends were similar but the magnitude of reductions varied

Using February 2020 as a baseline period, overall mobility in each city over time was quantified by the number of weekly trips in each city as a proportion of the average number of weekly trips in that city in February. General trends in mobility between February 2 –August 31, 2020 were similar across all cities in our dataset (Fig 1B). The number of weekly trips began to decline in early-to-mid March. Rates of travel were lowest in the week beginning March 15 in Columbus and between March 22 and April 12 in all other cities, with mobility in the Great Lakes and Northeast regions generally reaching lowest levels slightly earlier than elsewhere. The extent of these reductions in travel, however, were highly variable across the different cities in our dataset. The greatest reductions were seen on the West Coast (with minimum mobility rates of 58% of baseline on average), followed by the Northeast (61%) and Great Lakes regions (63%), while mobility decreased the least in the Midwest (70%). Within each of these four regions, reductions across cities were similar. On average, moderate reductions were seen in the South and Southeast (65% and 66%, respectively) but there was much more variation between cities in these regions, with minimum mobility rates lower than 60% of baseline in cities such as Miami, Austin and El Paso, compared to 76% in Houston and 74% in Charlotte. Mobility began to increase in all cities from mid-to-late April, largely levelling off by the beginning of June (or increasing at a slower rate in the Midwest and Northeast). Following this, rates of travel were relatively stable across all cities during the summer. Again, the amount of travel relative to baseline at this new stable level varied greatly. Broadly, cities and regions which saw larger reductions in travel between February–early April also had lower levels of travel during the summer period. For example, average mobility from June-August on the West Coast, where initial reductions were greatest, was 77% of baseline whereas in the Midwest average mobility had almost returned to baseline levels (97%) over the same period.

## Changes in mobility were similar across age groups but varied by distance and income levels

Relative changes in mobility over time by age group (18–34, 35–54, 55+ years) were remarkably similar across all cities in our dataset. In terms of absolute numbers of trips per person, across all cities subscribers in the youngest age group consistently travelled more than those in the middle age group throughout the study period. Similarly, in absolute terms the middle-aged subscribers consistently travelled more than the oldest age group across cities and over time. However, the relative changes in travel over time compared to the baseline period were very similar across all age groups. Where there were differences, younger age groups tended to travel more relative to baseline, both the youngest compared to middle age groups and middle compared to oldest age groups, with these differences often becoming slightly larger over time (as shown in the examples in Fig 2A, see S1 Fig for all cities). These differences were also often larger in cities in the South and Southeast, including Miami, Dallas, Jacksonville and San Antonio. The only reversal of this pattern was in Columbus and Fargo (both in the Midwest), where travel relative to baseline was often slightly higher in older age groups, although again absolute numbers of trips per person remained higher in younger age groups.

There was more evidence of variation in changes in mobility by distance. When routes (pairs of zip codes) were classified based on distance quartile, the trends seen in Atlanta and

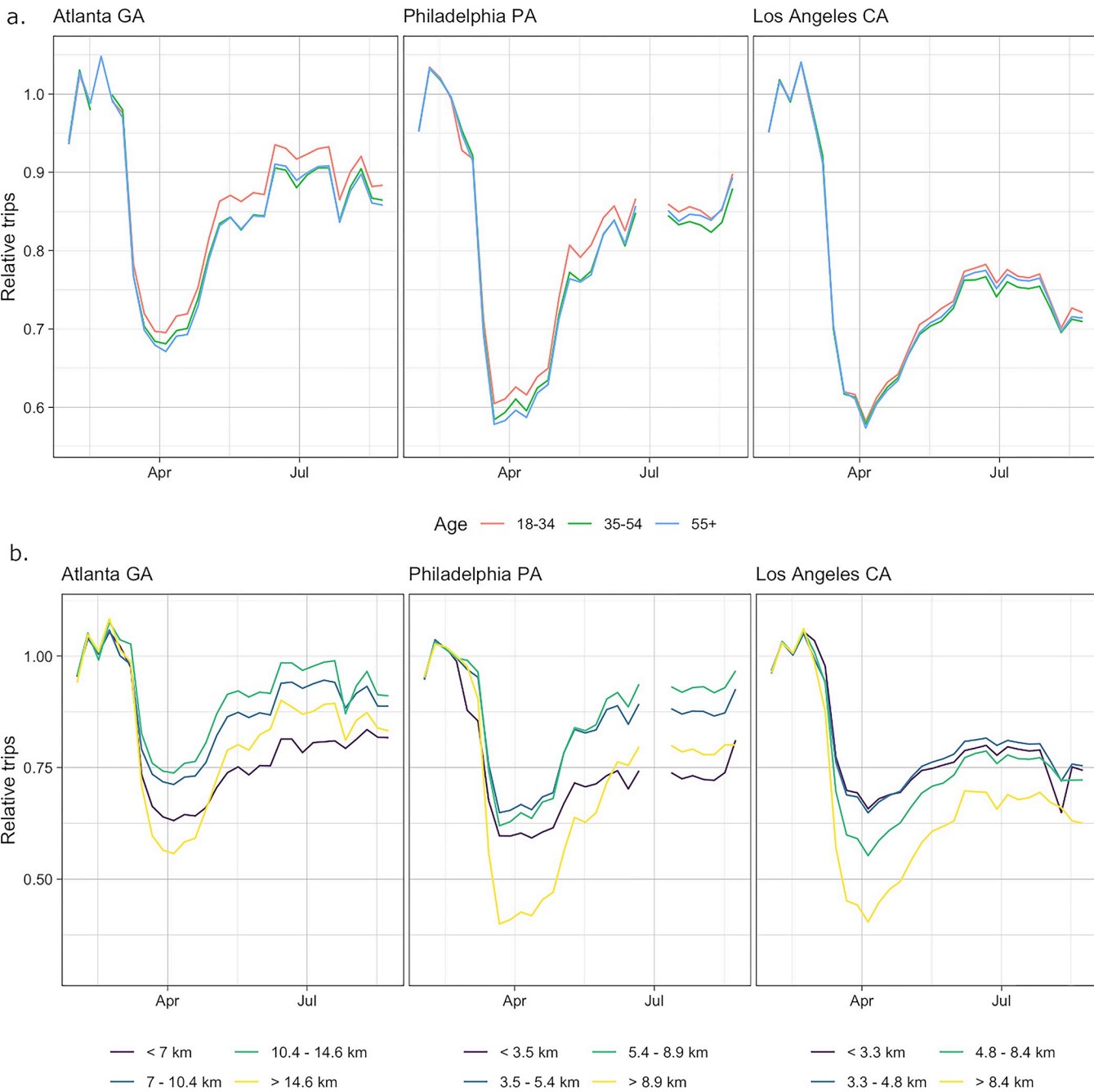

**Fig 2. Trips relative to baseline by age group and distance category in (left to right) Atlanta, Philadelphia and Los Angeles.** (a) Changes in relative travel by age were highly consistent in all three cities, particularly at the start of the pandemic when travel in all ages decreased at very similar rates. From late March onwards relative travel is slightly higher in the younger age groups but these differences are still small. (b) There was much more variation in mobility by distance. In Atlanta and Philadelphia, trips in the shortest and longest distance groups decreased significantly more initially than medium distance trips. From April, the rate of longest distance trips decreased faster than shortest distance from April (although both remained below medium distance trips). In Los Angeles, there was a more monotonic relationship where longer distance trips decreased at a faster rate than shorter ones (across all groups) and stayed at a lower level throughout.

Philadelphia (Fig 2B, see S2 Fig for all cities) typify those seen in many cities. During the initial decrease in travel beginning in early March, relative rates of longest distance travel reduced the most, followed by the shortest distance trips and then the two medium distance categories.

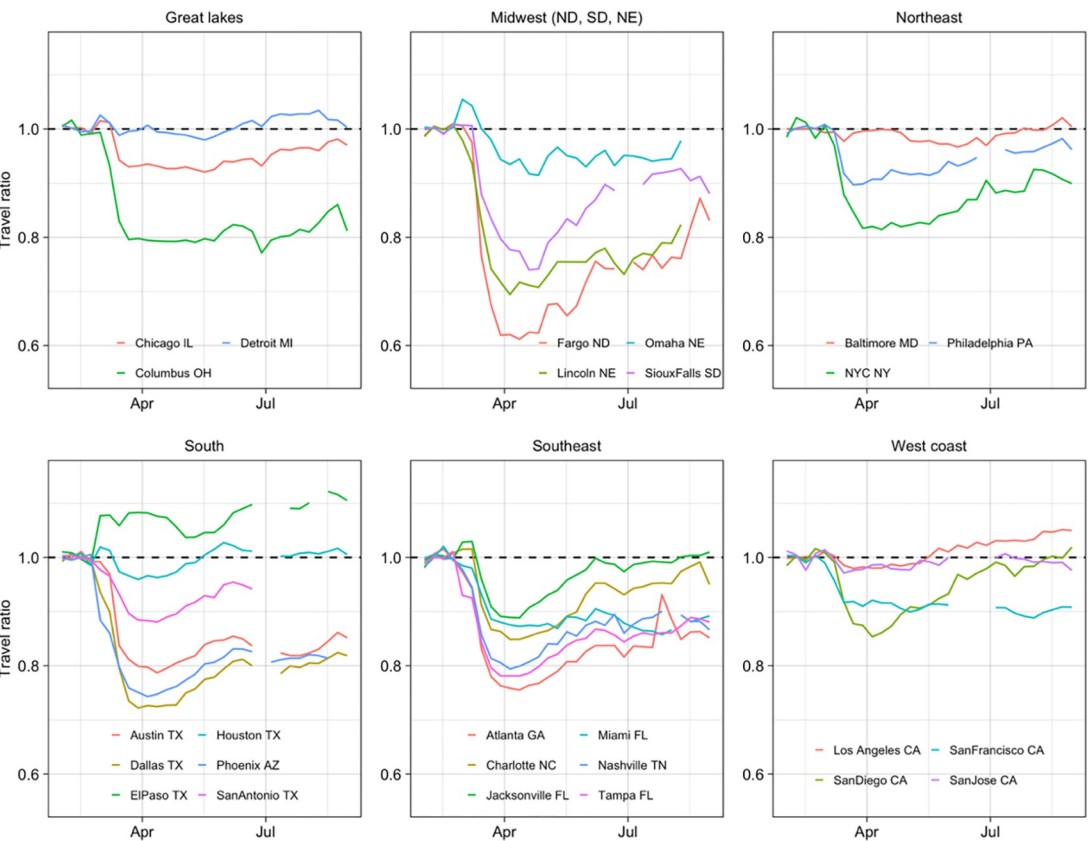

**Fig 3. Ratio of trips relative to baseline in zip codes with above and below the median proportion of higher-income subscribers.** Values below one indicate more travel relative to baseline in zip codes with fewer high income subscribers. As overall travel decreased from early March, there was evidence of greater reductions in trips from zip codes with more higher-income subscribers in most cities. This effect was not seen in Detroit, Baltimore, Houston, Los Angeles and San Jose. The opposite trend was seen in El Paso.

As overall travel increased in April, the longest distance trips increased faster than shortest trips so that by June the shortest distance trips were at a lower level relative to baseline than the longest distance trips (with the two medium distance groups remaining higher). However, there were also some cities (including Los Angeles, as shown in Fig 2B, San Diego and Omaha) where reductions in travel were monotonically associated with increasing distance across the whole study period (that is, longest trips decreased the most and shortest trips the least).

There was also evidence that changes in mobility depended on the income of subscribers in different zip codes. Fig 3 shows the ratio of numbers of outward trips, relative to baseline, in zip codes with above and below the median proportion of higher-income subscribers. In the large majority of cities, as travel decreased in early March this ratio also decreased, indicating that the outward trips from zip codes with higher proportions of higher-income subscribers were decreasing faster than elsewhere.

## Lowest rates of mobility and average summer mobility showed little association with NPIs or case rates at the city level

We compared average city-level mobility rates and state-level NPI stringency between March 15 –April 13, the time period during which all cities reached their lowest levels of mobility, and June 1 –August 30, when travel had generally stabilised. A similar comparison was made

between average mobility rates and COVID-19 cases per 100,000 (both at the city level) over the same time frames. Between March 15 –April 13, there was limited evidence of a negative association between both NPI stringency and case rates and travel (with correlations of -0.19 and -0.30 respectively, see S3 Fig). In both cases this relationship was weak and there were cities such as Phoenix and San Diego where relative travel was low despite low NPI stringency or case rates. While there was more evidence of negative associations between travel and both NPI stringency and case rates between June 1 –August 30, these associations were still fairly weak (correlations of -0.41 and -0.42). Cities where city or county-level restrictions were implemented in addition to state-level NPIs during this period are highlighted in S3 Fig. There is little evidence of a stronger association when taking these restrictions into account, with many cities with fairly high rates of travel (such as Dallas and Atlanta) included in those with additional restrictions.

## Bayesian analysis found long distance trips and higher incomes were associated with greater reductions in mobility

To better understand the effects of different variables on changing zip code-level mobility, we used two Bayesian models of mobility. These models allowed us to quantify the relative effects of different variables (age, distance, subscriber income, and case rates relative to the rest of the city) on the rate and extent to which travel between zip codes changed over time. They also allowed us to control for the median household income (of the general population, not limited to the subscribers in our dataset) in each zip code. Noise in the observation process and the censoring of values below 50 were accounted for in these models through the likelihood. The first model investigated the weekly rate of decrease in travel in each city between February 2 – April 4, 2020, the period of rapid decrease in mobility at the start of the pandemic in the US, and the effect of different variables on this rate. The variables with the largest and most consistent effects were distance between zip codes and the proportion of high-income subscribers in each zip code (Fig 4). These effects largely agreed with previous results, with mobility decreasing faster for longest distance trips and from zip codes with more higher-income subscribers. However, shortest distance trips were associated with a slower decrease in travel than other distances, in contrast to the trends seen in Fig 2. The effect of income was greatest in the South (except for El Paso and Phoenix where there was little to no effect) and Southeast, while being smallest on average in the Midwest and on the West Coast. The effect of distance was generally smaller in the Midwest and the overall trend was reversed in Fargo, where shorter trips decreased faster than longer ones. In most cities, relative case rates had little effect on travel. Where there was an effect, higher case rates were associated with faster decreases in travel, except for in Lincoln and San Diego. Age generally had a little effect on the rate of reduction in travel, although in the South, Southeast and Midwest older age groups were associated with slightly faster travel reductions.

The second model quantified the difference in zip code-level mobility between baseline levels (February 2020) and June 1 –August 31, 2020. Following the rapid decrease in travel observed during the first months of the pandemic and the subsequent increase, by this time period overall mobility had reached a relatively stable level. Characterising mobility at this time therefore gives us an insight into how changes in mobility at a fine spatial scale persisted beyond the initial months of the pandemic. Again, distance was an important factor here but unlike in the previous model, where we saw a monotonic relationship between distance and rate of decrease in travel, here both the longest and shortest distance categories were associated with less travel compared to medium distance trips (Fig 5). The effect of distance was generally smaller in the South, Southeast and Midwest than elsewhere. The relationship between travel

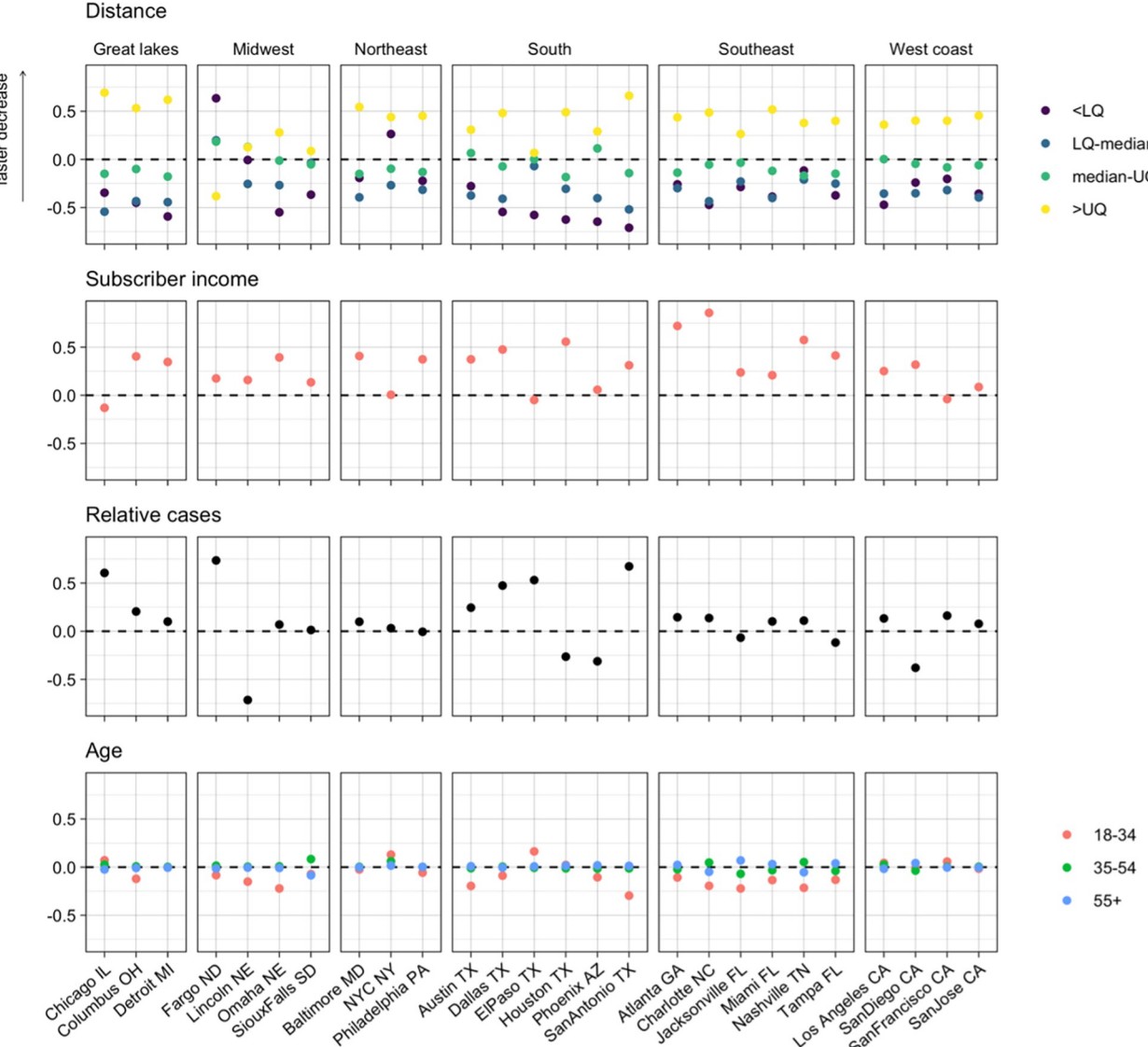

**Fig 4. Effects of distance, income, relative case rates, and age on rate of decrease in travel between February 1- April 3.** Distance and subscriber income had the most consistent effects, with longer trips and higher incomes associates with faster decreases in travel. The effects of relative case rates were inconsistent and effects of age were small. (LQ–lower quartile, UQ–upper quartile).

and income was very similar to that seen in the previous model, with higher incomes being associated with less travel and this effect being more prominent in the South (again except El Paso and Phoenix), Southeast and Midwest. There was an association between younger age groups and more travel in the South, Southeast and Northeast but elsewhere this effect was small.

It is worth noting that there was little association between subscriber incomes and the median household income (not limited to subscribers) at the zip code-level within each city. This meant that we could control for median household income without reducing our ability to infer the effect of subscriber income and that the effect of subscriber income can be interpreted as the effect of the individual's income on behaviour, rather than the income of the area where an individual lived.

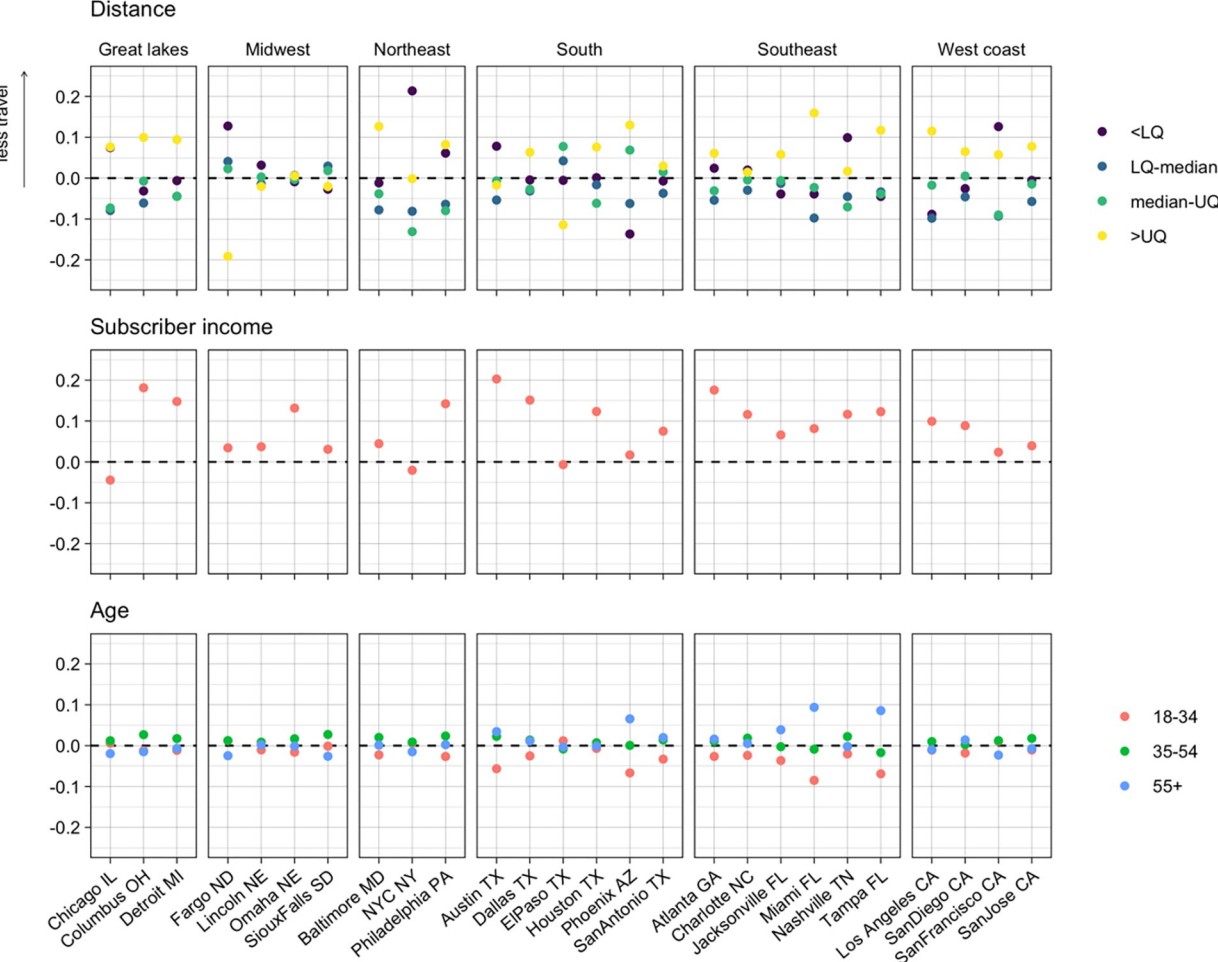

**Fig 5. Effects of distance, income and age on trip rates between June 1 –August 31 compared to baseline travel.** Longest and shortest distance trips were often associated with lower rates of travel, as were higher subscriber incomes. Younger age groups were associates with more travel in the South and Southeast. (LQ–lower quartile, UQ–upper quartile).

## Alternative gravity model approach overemphasised the effect of distance

Our model structure allowed for more complex baseline mobility patterns than traditional mobility models, such as the gravity model, which assume that mobility patterns are determined by distance and origin and destination population sizes. We fit an alternative model of the initial decrease in mobility using a gravity model structure for baseline travel to investigate whether the relationships learned using our original model could also be inferred with this more parametric model structure. The original model consistently greatly outperformed the gravity model structure in all cities, as measured using the Bayesian information criterion [46, 47] (see S5 Text for more details). The gravity model structure also produced markedly different results to the original model, with large positive values inferred for the coefficient for longest distance trips and small values for most other parameters, implying that only longest distance trips decreased significantly in the initial months of the pandemic.

## Discussion

Human mobility changed considerably in response to the COVID-19 pandemic. While many analyses have investigated changes in the overall volume of travel and its relationship to NPIs

and transmission rates, there have been fewer attempts to characterise how mobility patterns changed at a fine spatial scale. Understanding not just how much travel occurred at a given point in time, but what type of trips were being taken and by whom is crucial for understanding the effect of changes in mobility on transmission and the distribution of disease burden. Few analyses have made use of mobile device data as a source of very high spatial resolution mobility data in the context of the COVID-19 pandemic, as we have done here. Spatial aggregation can obscure patterns that exist at finer scales and effects of variables on mobility inferred at the aggregate level may not hold at finer scales or for individuals [48]. Similarly, mobility indices produced by companies such as Google, Apple, and Baidu [49–52] have been widely used [53–56] but pre-processing and aggregation of these data often limit the spatial detail available.

We analysed fine-scale, zip code-level mobile device data from 26 cities in the US to characterise changes in mobility patterns during the start of the COVID-19 pandemic (February 2 – August 31, 2020). General trends in total trip numbers were similar in all cities, with reductions and increases in travel occurring at similar times, but the magnitude of these changes varied considerably across locations. Our Bayesian modelling approach allowed us to quantify the relative effects of different factors on fine-scale mobility patterns while accounting for stochasticity and censoring, resulting in more robust and generalisable inferences than could be achieved by direct univariate comparisons alone. Our analysis revealed that changes in mobility were associated with trip distance and income level, but less associated with age group. During the initial decrease in travel in March, there was a monotonic relationship between trip distance and rate of decrease in almost all cities, with longer distance travel decreasing faster than shorter distance travel. This is similar to the decreases in long distance trips observed in Germany in response to the pandemic [35]. Over the summer distance was still strongly associated with rates of travel, however at this point in time trips in the shortest and longest distance quartiles were at lower levels relative to baseline compared to medium distance trips. This suggests two distinct patterns of mobility, during the initial decrease in travel and over the summer months, neither of which was a uniform decrease in mobility from baseline. It also highlights the persistence of changes in mobility beyond the initial shutdown, suggesting long-lasting structural changes in mobility patterns, as observed elsewhere [35]. Demographically, we found differences in mobility patterns by income groups supporting previous findings that lower income groups in the US were on average less able to comply with stay-at-home orders [57, 58]. However, we found little difference in the relative reductions in mobility across age groups, although higher baseline rates of travel in the younger age groups in all cities (both in the 18–34 group compared to 35–54 and 35–54 compared to 55+ years) meant that in an absolute sense younger age groups were still consistently travelling more than older age groups.

Interestingly, we saw little association between the average amount of travel relative to baseline in each city either around the beginning of April or during summer and the stringency of NPIs in effect over these same periods. There is evidence of voluntary modification of behaviour in the US in the early months of the pandemic regardless of the NPIs in place [11, 39] and, conversely, that lower income groups were on average less able to comply with stay-at-home orders [57, 58], which would both lessen the impact of NPIs. The geography of each city (including factors like the distance between households and schools, shops or workplaces) is also likely to have affected how interventions such as school or workplace closures translated into changes in travel patterns. Similarly, there was no clear relationship between average travel relative to baseline at the beginning of April or during summer and corresponding average COVID-19 case rates. Despite hypotheses that increased case rates may have resulted in individuals avoiding specific behaviours to minimize their risk of infection, we did not see this

relationship in the data. In particular, cities in the South and Southwest that had high levels of transmission during the summer (which peaked around mid-July) did not have noticeably lower rates of travel than elsewhere, perhaps suggesting that stable amounts of travel were able to help sustain transmission. In addition, within these cities there was little evidence of a reduction in travel between June and August (Fig 1B and S4 Fig), further highlighting the apparent disconnect between COVID-19 transmission and behaviour. We also found across all cities that local COVID-19 case rates being high relative to the rest of the city often had little association with reduction in travel at the zip code level in the initial months of the pandemic.

Some cities in the Midwest, particularly Lincoln and Sioux Falls, may have come the closest to returning to normal patterns of mobility in the summer, as in both overall number of trips had returned to baseline levels and there were no clear effects of distance, income, or age on differences in zip code-level travel relative to baseline. Comparing these cities to Columbus and Detroit, where overall numbers of trips also returned to near baseline levels but there were clear differences in zip code-level mobility patterns, again shows the importance of information on fine scale data for understanding mobility patterns.

There were several limitations to our work. Most of the subscribers with demographic data were in the oldest age group in all cities and the higher income groups were also consistently over-represented. As these demographic variables were adjusted for in the regression models, this is unlikely to substantially affect our zip-code level results. However, if older and higher-income individuals were more likely to reduce travel regardless of NPIs in effect then the association between NPIs and rates of travel at the city level may be greater in the general population than we found here. Furthermore, the low numbers of subscribers with data indicating they were in the lowest income group (earning less than $25,000 a year) meant that this group was combined with the next group (between $25,000 and $50,000 a year) in our analysis. Given the evidence of a relationship between low incomes and risk of COVID-19 infection, similar work including higher numbers of lower income subscribers is likely to be valuable. The age groups in our data were also fairly coarse, preventing us from investigating differences over smaller ranges (such as subscribers of typical college age compared to those in their early thirties) and potentially contributing to the lack of differences observed between age groups. In addition, the case data used was available at the county level, rather than by zip code, which may have reduced our power to detect an effect of local case rates on mobility. Finally, our baseline period (February 2020) was fairly short and covered a period of time where COVID-19 transmission was already occurring in the US. A longer and earlier baseline period would likely have allowed for a more reliable estimation of mobility patterns before the pandemic.

Human mobility underlies the spatial spread of infectious diseases. Understanding heterogenies in these patterns across space, time, and demographic groups is needed to inform predictions beyond just SARS-CoV-2. Despite substantial work in quantifying how human travel has changed during the COVID-19 pandemic to date, little work has been done to investigate fine-scale patterns of mobility and develop frameworks to model these changes and their potential impact on the risk of sustained epidemic transmission. Here, we presented an analysis of zip code level travel in 26 cities in the US between February 1 –August 31, 2020 and found consistent trends over time and effects of distance and income, but not age, on reductions in mobility, despite much heterogeneity in the magnitude of these reductions across cities. There was little evidence of a relationship across cities between mobility levels relative to baseline from mid-March–early April and either NPI stringency or case rates. Similarly, there was little association between these variables during the summer. This highlights the importance of direct measurements of behaviour like the mobile device data analysed here for understanding mobility patterns during the pandemic.

## Supporting information

**S1 Data. List of zip codes used.**
(CSV)

**S2 Data. Weeks removed due to data loss.**
(CSV)

**S1 Text. Model results with alternative distance quartiles.**
(PDF)

**S2 Text. Demographic distributions [60, 61].**
(PDF)

**S3 Text. Prior distributions.**
(PDF)

**S4 Text. Modification to Poisson likelihood for censored observations.**
(PDF)

**S5 Text. Comparison to gravity model.**
(PDF)

**S1 Table. Summary of all models used.**
(PDF)

**S2 Table. Proportion of subscribers in each city.**
(PDF)

**S1 Fig. Trips relative to baseline by age group in each city.**
(PDF)

**S2 Fig. Relative trips by distance quartile for each city.**
(PDF)

**S3 Fig.** Relationship between relative travel and NPI stringency (left) and case rates (right) at the city level between (a) March 15 –April 12 2020 and (b) June 1 –August 30 2020. NPI stringency was at the state level. Cities with some significant additional NPIs in place (beyond the state-level policies) between June 1 –August 30 are highlighted in red. (a) There was little association between travel and either NPI stringency or case rates between March 15 –April 12. (b) When taking into account cities with additional NPIs, there was some evidence of a weak association between NPI stringency and travel between June 1 –August 30. There was little association between travel and case rates over this period.
(PDF)

**S4 Fig. Daily trips relative to baseline in cities in the South (excluding San Antonio) between June 1 –August 31, with days with data loss removed.** Using daily trips reduces the amount of missingness as in the weeks with data loss there was only data loss in some (not all) days. With the exception of Phoenix, where travel decreased during June from above baseline levels to around 90% of baseline, there is little evidence of a decreasing mobility during this time frame. San Antonio is not included as there was a similar amount of data loss in weekly and daily data.
(PDF)

## Author Contributions

**Conceptualization:** Rohan Arambepola, Kathryn L. Schaber, Derek A. T. Cummings, Amy Wesolowski.

**Data curation:** Rohan Arambepola, Kathryn L. Schaber, Catherine Schluth, Amy Wesolowski.

**Formal analysis:** Rohan Arambepola, Kathryn L. Schaber, Catherine Schluth.

**Funding acquisition:** Amy Wesolowski.

**Methodology:** Rohan Arambepola, Kathryn L. Schaber, Catherine Schluth, Angkana T. Huang, Alain B. Labrique, Shruti H. Mehta, Sunil S. Solomon, Derek A. T. Cummings.

**Supervision:** Amy Wesolowski.

**Visualization:** Rohan Arambepola.

**Writing – original draft:** Rohan Arambepola, Kathryn L. Schaber, Catherine Schluth, Angkana T. Huang, Alain B. Labrique, Shruti H. Mehta, Sunil S. Solomon, Derek A. T. Cummings, Amy Wesolowski.

**Writing – review & editing:** Rohan Arambepola, Kathryn L. Schaber, Catherine Schluth, Angkana T. Huang, Alain B. Labrique, Shruti H. Mehta, Sunil S. Solomon, Derek A. T. Cummings, Amy Wesolowski.

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
