## [Decision Letter · Decision Letter 0]

18 Jan 2023

PGPH-D-22-01784

Fine scale human mobility changes in 26 US cities in 2020 in response to the COVID-19 pandemic were associated with distance and income

Dear Dr. arambepola,

Thank you for submitting your manuscript to PLOS Global Public Health. After careful consideration, we feel that it has merit but does not fully meet PLOS Global Public Health’s publication criteria as it currently stands. Therefore, we invite you to submit a revised version of the manuscript that addresses the points raised during the review process.

We look forward to receiving your revised manuscript.

Kind regards,

Max Carlos Ramírez-Soto, BSc, MPH, FRSPH, MACE

Academic Editor

Journal Requirements:

2. Please send a completed 'Competing Interests' statement, including any COIs declared by your co-authors. If you have no competing interests to declare, please state "The authors have declared that no competing interests exist". Otherwise please declare all competing interests beginning with the statement "I have read the journal's policy and the authors of this manuscript have the following competing interests:"

3. Please amend your detailed Financial Disclosure statement. This is published with the article. It must therefore be completed in full sentences and contain the exact wording you wish to be published.

a. State what role the funders took in the study. If the funders had no role in your study, please state: “The funders had no role in study design, data collection and analysis, decision to publish, or preparation of the manuscript.”

b. If any authors received a salary from any of your funders, please state which authors and which funders.

4. Please provide separate figure files in .tif or .eps format only and remove any figures embedded in your manuscript file. Please also ensure that all files are under our size limit of 10MB.

5. We noticed that you used "data not shown and unpublished data" in the manuscript. We do not allow these references, as the PLOS data access policy requires that all data be either published with the manuscript or made available in a publicly accessible database. Please amend the supplementary material to include the referenced data or remove the references.

6. We have noticed that you have uploaded Supporting Information files, but you have not included a list of legends. Please add a full list of legends for your Supporting Information files after the references list.

7. Fig 1: please (a) provide a direct link to the base layer of the map (i.e., the country or region border shape) and ensure this is also included in the figure legend; and (b) provide a link to the terms of use / license information for the base layer image or shapefile. We cannot publish proprietary or copyrighted maps (e.g. Google Maps, Mapquest) and the terms of use for your map base layer must be compatible with our CC-BY 4.0 license. 

Additional Editor Comments (if provided):

Reviewers' comments:

Reviewer's Responses to Questions

**Comments to the Author**

1. Does this manuscript meet PLOS Global Public Health’s publication criteria? Is the manuscript technically sound, and do the data support the conclusions? The manuscript must describe methodologically and ethically rigorous research with conclusions that are appropriately drawn based on the data presented.

Reviewer #1: Yes

Reviewer #2: Yes

2. Has the statistical analysis been performed appropriately and rigorously?

Reviewer #1: I don't know

Reviewer #2: Yes

3. Have the authors made all data underlying the findings in their manuscript fully available (please refer to the Data Availability Statement at the start of the manuscript PDF file)?

Reviewer #1: No

Reviewer #2: Yes

4. Is the manuscript presented in an intelligible fashion and written in standard English?

Reviewer #1: Yes

Reviewer #2: Yes

5. Review Comments to the Author

Reviewer #1: Thank you for the opportunity to review this manuscript. In this paper, Arambepola et al used aggregated US mobile phone operator data to understand the movement patterns across zip codes in 26 cities in the US. Using February 2020 for baseline rates of movement they found that there was overall decrease in mobility early in the pandemic with differences in magnitude varying by geography. Changes in movement across zip codes was associated with trip distance and income level. Age-group associated changes were similar across the different cities. There have been many scientific papers published now on the changes of respiratory virus activity associated with NPI implementation during the pandemic but very few to my knowledge that actually quantify the changes the mobility pattern (one aspect of community level NPI). I believe this manuscript to be of interest to the readers of PLOS Global Public Health. I have some minor considerations below for the authors as they revise their manuscript. Suggest including line numbers for ease of reference during revision.

1. As I started reading the introduction, I found myself asking how the authors defined the scale of mobility in this study. Obviously, mobility can be defined at a number of levels from international travel, interstate and even within neighborhoods and homes. It required that I read the methods before I understood that what this study was about intra-city/zip code travel. The abstract briefly represents zip code-level travel but this is “within the city” travel and not between zip codes of different states. Suggest defining this clearly and early in the title, abstract and introduction.

2. Introduction – This manuscript is packed full of information which I appreciate, however, for the introduction I might suggest hitting the higher level points and reserve more detailed content for the discussion. For example, I think the 3rd paragraph (beginning with “in response to the pandemic,…) could supplement discussions in the discussion rather than being situated in the introduction. Further more, I might suggest that the 4th paragraph be limited to 2 sentences to briefly discuss the objectives and let the rest of the paper go into more methodologic detail. This will help ensure that the messaging is concise and not overly repeated.

3. The conventional format for many scientific articles has methods prior to presentation of results. In some cases, the order might be enhanced by having results before methods but I think in this case, I would suggest that the methods come before the results. Much of the framework for understanding the results required detailed understanding of the methods and I think the order here of having methods before results is important. Suggest that the authors consider this formatting change.

4. Can the authors describe how the 26 city data were chosen? Why these 26 cities? Or were these the only ones that the mobile providers gave?

5. Methods – I believe there to be a typo in the age group description in the methods. The age-group 44-54 is missing.

6. Methods (Demographic Data) – The sentence beginning “subscribers in the dataset with available income and age data…” is results and not methods.

7. Results – Are the authors able to provide overall description of the dataset? How many mobile providers submitted data? How many unique cell phone users? What was the proportion breakdown by city/state?

Reviewer #2: Article title reviewed: Fine scale human mobility changes in 26 US cities in 2020 in response to the COVID- 19 pandemic were associated with distance and income.

Output of the review (This review has been also uploded as an attachment):

1. The manuscript claims to define a modelling framework of changes in mobility on a fine spatial scale in the COVID-19 pandemic context.

2. Is this article a clinical case report, a proposal or hypothesis paper, a protocol, a letter, an essay, clinical practice guidelines, a monograph?

The response is No. This article can be classified as primary literature: in the absence of vaccines against COVID-19, approaches to reduce viral transmission are behavioral, involving policies that reduce human mobility. Thus, Covid-19 has quickly become an important use case to model aggregate human mobility patterns in response to large scale mobility reduction. Measurement of flows and changes in human mobility contributes to a better understanding of how infectious diseases spread to new places, and enables more accurate estimation of the impact of behavioral changes on viral transmission rates. The major specify of this article is the mobility on a fine spatial scale.

3. This article has been preprinted under doi: 10.1101/2022.11.04.22281943. It is available: www.medrxiv.org/content/10.1101/2022.11.04.22281943v1

4. The investigation was carried out according to methods recognized as being rigorous based on proven mathematical theories, relevant software and standard material:

• Bayesian theory, Laplace approximations, Poisson Model.

• R software.

• Results of the 2019 American Community Survey (ACS).

• COVID-19 Data Repository by the Center for Systems Science and Engineering (CSSE) at Johns Hopkins University using the COVIDcast Epidata.

• Keystone COVID-19 Intervention Dataset.

• Hikma Health COVID-19 Policy Dataset.

5. Regarding ethical and respect for participants private life aspects, this study has been conducted according to the ethical standards accepted generally into the scientific community with possibility of verification. For examples:

• the consent for participants is waived by the Institutional Review Board of Johns Hopkins Bloomberg School of Public Health.

• Mobile device data: any values below 50 were not supplied to protect subscriber anonymity.

• Demographic data: sociodemographic information for groups with fewer than 50 members was not included to protect the privacy of subscribers.

Main recommendations:

1. Methods section and sub-section Mobile device data: Some weeks in certain cities were excluded from the analysis due to data loss during the collection process (see supplementary material). In general, what was the technique used to process the missing data in this study? That must appear clearly in the article.

2. Similarly, the sampling determination method and the data analysis technique must be described and presented clearly as special sub-sections in the Methods section.

6. PLOS authors have the option to publish the peer review history of their article (what does this mean?). If published, this will include your full peer review and any attached files.

**Do you want your identity to be public for this peer review?** For information about this choice, including consent withdrawal, please see our Privacy Policy.

Reviewer #1: No

Reviewer #2: **Yes: **Lazare M BOUNGOU

---

## [Decision Letter · Decision Letter 1]

20 Jun 2023

Fine scale human mobility changes within 26 US cities in 2020 in response to the COVID-19 pandemic were associated with distance and income

PGPH-D-22-01784R1

Dear Dr arambepola,

We are pleased to inform you that your manuscript 'Fine scale human mobility changes within 26 US cities in 2020 in response to the COVID-19 pandemic were associated with distance and income' has been provisionally accepted for publication in PLOS Global Public Health.

Best regards,

Max Carlos Ramírez-Soto, BSc, MPH, FRSPH, MACE

Academic Editor

The authors have properly reviewed the manuscript. Therefore, I suggest accepting the manuscript.

Reviewer Comments (if any, and for reference):

Reviewer's Responses to Questions

**Comments to the Author**

1. If the authors have adequately addressed your comments raised in a previous round of review and you feel that this manuscript is now acceptable for publication, you may indicate that here to bypass the “Comments to the Author” section, enter your conflict of interest statement in the “Confidential to Editor” section, and submit your "Accept" recommendation.

Reviewer #1: All comments have been addressed

Reviewer #2: All comments have been addressed

2. Does this manuscript meet PLOS Global Public Health’s publication criteria? Is the manuscript technically sound, and do the data support the conclusions? The manuscript must describe methodologically and ethically rigorous research with conclusions that are appropriately drawn based on the data presented.

Reviewer #1: Yes

Reviewer #2: Yes

3. Has the statistical analysis been performed appropriately and rigorously?

Reviewer #1: I don't know

Reviewer #2: Yes

4. Have the authors made all data underlying the findings in their manuscript fully available (please refer to the Data Availability Statement at the start of the manuscript PDF file)?

Reviewer #1: Yes

Reviewer #2: Yes

5. Is the manuscript presented in an intelligible fashion and written in standard English?

Reviewer #1: Yes

Reviewer #2: Yes

6. Review Comments to the Author

Reviewer #1: Thank you for taking the time to address my comments, questions and concerns. I have no further suggestions at this time.

Reviewer #2: No comment.

7. PLOS authors have the option to publish the peer review history of their article (what does this mean?). If published, this will include your full peer review and any attached files.

**Do you want your identity to be public for this peer review?** For information about this choice, including consent withdrawal, please see our Privacy Policy.

Reviewer #1: No

Reviewer #2: **Yes: **Lazare MBOUNGOU
